# Bilirubin Links *HO-1* and *UGT1A1*28* Gene Polymorphisms to Predict Cardiovascular Outcome in Patients Receiving Maintenance Hemodialysis

**DOI:** 10.3390/antiox10091403

**Published:** 2021-08-31

**Authors:** Yang Ho, Tzen-Wen Chen, Tung-Po Huang, Ying-Hwa Chen, Der-Cherng Tarng

**Affiliations:** 1Division of Nephrology, Department of Medicine, Taipei Veterans General Hospital, Taipei 11217, Taiwan; yho@vghtpe.gov.tw; 2Faculty of Medicine, School of Medicine, National Yang Ming Chiao Tung University, Taipei 11221, Taiwan; 3Division of Nephrology, Wei Gong Memorial Hospital, Miaoli 35159, Taiwan; twchen@weigong.org.tw (T.-W.C.); tphuang12@gmail.com (T.-P.H.); 4Division of Cardiology, Department of Medicine, Taipei Veterans General Hospital, Taipei 11217, Taiwan; 5Center for Intelligent Drug Systems and Smart Bio-Devices (IDS2B), Hsinchu 30010, Taiwan; 6Department and Institute of Physiology, National Yang Ming Chiao Tung University, Taipei 11221, Taiwan; 7Department of Medicine, Taipei Veterans General Hospital, Taipei 11217, Taiwan

**Keywords:** cardiovascular events, *heme oxygenase-1*, hemodialysis, mortality, *UGT1A1*

## Abstract

Serum bilirubin levels, which are determined by a complex interplay of various enzymes, including heme oxygenase-1 (HO-1) and uridine diphosphate–glucuronosyl transferase (UGT1A1), may be protective against progression of cardiovascular disease (CVD) in hemodialysis patients. However, the combined effect of *HO-1* and *UGT1A1*2**8* gene polymorphisms on CVD outcomes among hemodialysis patients is still unknown. This retrospective study enrolled 1080 prevalent hemodialysis patients and the combined genetic polymorphisms of *HO-1* and *UGT1A1* on serum bilirubin were analyzed. Endpoints were CVD events and all-cause mortality. Mean serum bilirubin was highest in patients with S/S + S/L of the *HO-1* promoter and *UGT1A1* 7/7 genotypes (Group 1), intermediate in those with S/S + S/L of the *HO-1* promoter and *UGT1A1* 7/6 + 6/6 genotypes (Group 2), and lowest in the carriers with the L/L *HO-1* promoter and *UGT1A1* 7/6 + 6/6 genotypes (Group 3) (*p* < 0.001). During a median follow-up of 50 months, 433 patients developed CVD. Compared with patients in Group 3, individuals among Groups 1 and 2 had significantly lower risks for CVD events (adjusted hazard ratios (aHRs) of 0.35 for Group 1 and 0.63 for Group 2), respectively. Compared with the lower bilirubin tertile, the aHRs were 0.72 for the middle tertile and 0.40 for the upper tertile for CVD events. We summarized that serum bilirubin as well as *HO-1* and *UGT1A1* gene polymorphisms were associated with CVD among patients receiving chronic hemodialysis.

## 1. Introduction

Despite the technological advances in hemodialysis (HD) procedures and medical support in recent years, age-standardized cardiovascular mortality among dialysis patients is 8.8-fold higher than in the general population [1]. Due to vascular inflammation combined with oxidative stress [2,3], end-stage renal disease (ESRD) patients are at higher risk for development of cardiovascular disease (CVD) and even mortality. Although ESRD patients have a plethora of traditional and non-traditional risk factors, the underlying mechanisms for the development of CVD morbidity and mortality are still unknown. Accordingly, it is crucial to identify a new prognostic marker of CVD for both appropriate measurement and to permit better identification of high-risk groups of ESRD patients undergoing regular HD.

Bilirubin has both antioxidant [4,5] and anti-inflammatory [6,7] properties. The antioxidant and antiatherogenic effects of bilirubin are thought to result from its ability to inhibit the oxidation of low-density lipoprotein (LDL) and other lipids [8,9], scavenge oxygen radicals, and counteract oxidative stress [10,11]. Bilirubin has consistently been associated with protection against the development of coronary heart disease, peripheral vascular disease, and stroke [12,13,14,15]. Bilirubin is engendered through the catalysis of the heme degradation by heme oxygenase (HO) [16]. Further processing of bilirubin occurs in hepatocytes, where the uridine diphosphate–glucuronosyl transferase (UGT1A1) conjugates the lipid-soluble bilirubin to a water-soluble form for excretion. Serum bilirubin varies between individuals and can increase moderately through either induction of HO activity or partial inhibition of UGT1A1. 

The HO enzyme system, the rate-limiting enzyme in heme degradation, comprises two isoforms, HO-1 and HO-2 [17]. While HO-2 is expressed constitutively, HO-1 expression is induced by numerous stimuli that impose cellular oxidative stress [18]. The human *HO-1* gene has been mapped to chromosome *22q12* [19], and the number of guanosine thymidine dinucleotide repeats [(GT)n] in the *HO-1* gene microsatellite promoter is inversely associated with HO-1 mRNA levels and enzyme activity [20]. Our previous study demonstrated that chronic HD patients with longer lengths of (GT)n in the *HO-1* gene promoter exhibit higher inflammation and oxidative stress and have higher risk of long-term CVD events and mortality [21]. 

The human *UGT1A1* gene has been mapped to chromosome *2q37*. A common cause of decreased UGT1A1 activity is the insertion of a TA in the TATAA box in the promoter region of the *UGT1A1* gene [22,23,24]. Individuals homozygous for seven repeats (7/7) have higher levels of serum bilirubin than heterozygotes (7/6) or those with the wild type of six repeats (6/6) [24,25,26]. We observed that the *UGT1A1* gene polymorphism had strong effects on bilirubin levels and the 7/7 genotype might have an important effect on reducing CVD events and death in HD patients [27]. Accordingly, in this study, we attempted to clarify if there is a combined effect of the polymorphisms of the *HO-1* and *UGT1A1* genes on serum bilirubin and determine its impact on CVD events and all-cause mortality among chronic HD patients.

## 2. Materials and Methods

### 2.1. Study Population

The study was approved by the Institutional Review Board of Taipei Veterans General Hospital, and we performed a retrospective chart review to recruit patients receiving regular HD in nine dialysis centers from December 2016 to January 2021. The informed consent form was waived since our study was conducted retrospectively using human body materials and data and the subjects cannot be identified, and furthermore the research project did not involve personal privacy and commercial interests. The inclusion criteria were patients with age of >20 years, dialysis vintage for ≥6 months, dialysis using a standard bicarbonate dialysate, and genotyping of the number of (GT)n in the *HO-1* gene microsatellite promoter and the TATAA box in the promoter region of the *UGT1A1* gene. Patients were excluded if they had a weekly dialysis time less than 12 h, inadequate dialysis with urea Kt/V less than 1.2, malignancy, frequent episodes of infectious disease or sepsis, and hepatobiliary disease within 3 months prior to December 2016. All patients in the database system were Taiwanese and their ethnic backgrounds were similar. Therefore, the statistical artifacts caused by population stratification can be ruled out [28].

### 2.2. Clinical and Laboratory Data Collection

The database system was complemented by demographic data and clinical assessments of body weight, body mass index, and blood pressure (BP). Hypertension was defined as a measured systolic BP greater than 140 mmHg, a diastolic BP greater than 90 mmHg, and/or use of antihypertensive medications. Diabetes was diagnosed on the basis of the World Health Organization criteria. The presence of CVD was defined as a medical history and clinical findings of congestive heart failure and coronary artery, cerebrovascular, and/or peripheral vascular disease. 

Laboratory data were the mean value of each parameter within 3 months before December 2016. Venous blood was drawn from HD patients who had fasted overnight at a midweek dialysis session. Laboratory data included albumin, iron, total cholesterol, triglyceride, HDL cholesterol, LDL cholesterol, urea, creatinine in the serum, total iron-binding capacity, serum ferritin, and high-sensitivity C-reactive protein (hs-CRP). Transferrin saturation was calculated as the ratio of serum iron to TIBC and presented as a percentage. Plasma malondialdehyde was determined with a thiobarbituric acid test. The adducts consisting of two molecules of thiobarbituric acid were separated by the HPLC method described by Nielsen et al [29]. Total serum bilirubin was measured using the metavanadate oxidation method (Wako Pure Chemical Industries, Ltd., Osaka, Japan). The adequacy of dialysis was estimated by measuring the midweek urea clearance (Kt/V) using the standard method [30].

### 2.3. Genotyping Methods

Two genotypes have been measured in the database system since 2006. The 5′-flanking region containing (GT)n repeats of the *HO-1* gene was amplified by the PCR with a 5-carboxyfluorescein-labeled sense primer (5′-AGAGCCTGCAGCTTCTCAGA-39) and an antisense primer (59-ACAAAGTCTGGCCATAGGAC-3′), following a previously published procedure [17]. The present study selected 27 GT repeats as a cutoff to classify the participants for allele typing; the proportion of allele frequencies with less than 27 GT repeats was approximately 50%, and the cutoff value was consistent with previously published studies from the literature [31,32,33]. In accordance with each of the *HO-1* promoter alleles, the patients were categorized into L/L, L/S, or S/S genotypes. 

Genotyping of the *UGT1A1* promoter TA-repeat polymorphism in the TATA box at position -53 was performed using the ABI 3130 x l sequencing system as recently described in detail [34]. PCR was performed with a 5-FAM (carboxyfluorescein)-labeled forward primer (5′-CACGTGACACAGTCAAAC-3′) and an unlabeled reverse primer (5′-CAACAGTATCTTCCCAGC-3′). The PCR products were sequenced to determine the number of TA repeats over the promoter of the *UGT1A1* gene. Patients were categorized into homozygous genotypes for seven repeats (7/7), heterozygous genotypes (7/6), or those with the wild type of six repeats (6/6).

### 2.4. Outcome Data Collection

Study outcomes were CVD events and all-cause mortality. The composite CVD events included fatal and nonfatal myocardial infarction and stroke, congestive heart failure, peripheral artery disease, and sudden death. The all-cause mortality included death related to CVD events, infection, sepsis, malignancy, gastrointestinal bleeding, chronic obstructive lung disease, and cachexia.

### 2.5. Statistical Analysis

Baseline descriptive variables were expressed as percentages for categorical data, mean values ± SD for continuous data with a normal distribution, and medians and interquartile ranges for continuous data without a normal distribution. In our previous studies, we found that HD patients with the 7/7 genotype of the *UGT1A1* exhibited higher bilirubin levels as compared to the 7/6 and 6/6 genotypes [27]. Moreover, those with the 7/7 genotype of the *UGT1A1* promoter were associated with the lowest risk for CVD events and all-cause death as compared to the patients with the 6/6 and 6/7 genotypes. In contrast, HD patients with the L/L genotype of the HO-1 gene promoter had higher oxidative stress with increased plasma MDA and C-reactive protein levels as compared to those with S/L and S/S genotypes. Furthermore, patients with L/L were associated with the highest CVD events and all-cause mortality as compared to those with L/S and S/S genotypes. To examine the combination effects of three genotypes of the *HO-1* gene promoter (S/S, S/L, and L/L) with three genotypes of the *UGT1A1* gene (7/7, 7/6, and 6/6) on serum bilirubin, three genetic models including S/S + S/L with 7/7 (Group 1); L/L with 7/7 and S/S + S/L with 7/6 + 6/6 (Group 2); and L/L with 7/6 + 6/6 (Group 3) were constructed (Figure 1). Potential differences among the three patient groups were assessed with ANOVA for normally distributed data, the Kruskal–Wallis test for nonnormally distributed data, or the Pearson chi-squared test for categorical variables. Linear regression was used to test differences in mean levels of serum bilirubin and plasma malondialdehyde among three genetic models. A multivariate Cox regression model was used to estimate the hazard ratios of composite CVD events and all-cause mortality in relation to serum bilirubin and three genetic models. The analysis was adjusted for age, sex, smoking status, diabetes, prior cardiovascular disease, body mass index, total cholesterol, systolic blood pressure, hemodialysis duration, serum albumin, hs-CRP, and hemoglobin. Statistical analyses were performed using the computer software SPSS, version 27.0 (SPSS Inc., Chicago, IL, USA). All *p* values were two-tailed. *p* values less than 0.05 were considered statistically significant.

## 3. Results

### 3.1. Baseline Characteristics of Patients

Initially, 1272 HD patients were screened and 1080 patients (552 men and 528 women; mean age of 59 years) were enrolled from the database system. Their HD vintage (median) before the enrollment was 50 months. The baseline demographic characteristics and traditional and dialysis-related risk factors of the study population stratified into three groups are presented in Table 1. There was an even distribution of characteristics among the three patient groups including age, gender, smoking history, diabetes, hypertension, prior CVD, LDL-cholesterol, urea Kt/V, BMI, dose of epoetin, and duration of HD.

### 3.2. Serum Bilirubin and Oxidative Stress in HD Patients

Mean serum bilirubin in the patients of Group 1 (1.08 ± 0.32 mg/dL; *p* = 0.009) and Group 2 (0.81 ± 0.16 mg/dL; *p* = 0.045) were significantly higher as compared to those in Group 3 (0.66 ± 0.28 mg/dL), respectively (Figure 2A). The bilirubin level of Group 1 was 63% higher than that of Group 3. Plasma malondialdehyde levels in the patients of Group 1 (1.62 ± 0.48 mg/dL; *p* = 0.007) and Group 2 (1.78 ± 0.74 mg/dL; *p* = 0.012) were significantly lower than those in Group 3 (2.87 ± 0.65 mg/dL), respectively (Figure 2B). The results suggested that the patients in Group 3 exhibited the highest oxidative stress. 

### 3.3. CVD Events and All-Cause Mortality

At the end of January 2021, 73 patients received a kidney transplant, 12 patients were transferred to peritoneal dialysis, and 151 patients were transferred to other dialysis units. The frequency distribution of (GT)n of the *HO-1* promoter and genotypes of the *UGT1A1* promoter in the censored patients was similar to the frequency distribution in non-censored patients. Overall, 433 patients developed CVD events. Of the 307 patients who died, 139 (45.3%) patients died from CVD-related causes.

Table 2 displays the relative risk for CVD events and all-cause mortality in the HD patients stratified by the tertiles of bilirubin and three combined genotype groups, respectively. Multivariate analysis by Cox regression model indicated that patients in Group 1 had approximately one-third the risk for CVD events (adjusted hazard ratio (aHR) of 0.35 (95% confidence interval (CI), 0.15 to 0.89); *p* = 0.024) and all-cause mortality (aHR 0.40 (95% CI, 0.16 to 0.99), *p* = 0.049) compared with those in Group 3, respectively. Individuals in Group 2 also had a significantly lower risk for CVD events (aHR 0.63 (95% CI, 0.50 to 0.83), *p* < 0.001) and a trend toward reducing all-cause mortality (aHR 0.81 (95% CI, 0.68 to 1.01), *p* = 0.060). Compared with patients with lower tertiles of bilirubin, middle tertile (aHR 0.72 (95% CI, 0.50 to 0.96), *p* = 0.045) and upper tertile (aHR 0.40 (95%, 0.30 to 0.53), *p* < 0.001) patients had lower risks for CVD events.

## 4. Discussion

In the present study, we first report that serum bilirubin was modulated by the synergistic effect of the *HO-1* and *UGT1A1* gene polymorphisms. Group 1 patients with the combination of the L/L genotype with the *HO-1* gene and the 7/6 + 6/6 genotypes with the *UGT1A1* gene had the lowest serum bilirubin levels. The number of guanosine thymidine dinucleotide repeats ((GT)n) in the *HO-1* gene microsatellite promoter is inversely associated with HO-1 mRNA levels and enzyme activity [20]. Taha, H. et al. further found that HUVEC cells carrying the S allele survived better under oxidative stress and cell proliferation was more efficient in response to vascular endothelial growth factor A. Moreover, the presence of the S allele was associated with lower production of some proinflammatory mediators [35]. In the present study, the allelic distribution ranged from 16 to 39 GT repeats, with 23 and 30 GT repeats being the two most common alleles. We selected 27 GT repeats as a cutoff to classify the HD patients for allele typing, and short repeats with less than 27 GT repeats were designated as S allele and long repeats with at least 27 GT repeats were designated as L allele. Our previous study showed that serum CRP and plasma malondialdehyde levels were higher in patients with the L/L genotype as compared to those with the S/S genotype. Our findings suggested that the L/L genotype was associated with the status of higher inflammation and oxidative stress [21]. Intriguingly, in Group 1, there was a relatively lower proportion of patients with hypertension, and this might have been associated with lower oxidative stress and inflammation, as well as lower endothelial dysfunction, in the presence of one or two S alleles in HOMX1 expression [35].

The traditional CVD risk factors, like diabetes, hypertension, hyperlipidemia, obesity, and physical inactivity, are more prevalent among ESRD patients. However, the abovementioned factors seem to underrate accelerated CVD in ESRD patients receiving regular HD. Oxidant stress and inflammation play a pivotal role in the development and progression of CVD and the subsequent complications [36,37]. HD patients may be confronted with a loss of some low-molecular-weight factors in plasma, like vitamins A, C, and E [38,39], which normally attenuate inflammation by neutralizing reactive oxygen species (ROS) [40]. ROS are increased during HD through the direct contact of blood components with artificial biocompatible dialysis membranes [41]. The imbalance between antioxidants and pro-oxidants engenders oxidative stress that exacerbates the inflammatory state already present in these patients.

Bilirubin has both antioxidant [4,5] and anti-inflammatory [6,7] properties. A higher concentration of total serum bilirubin was associated with a lower risk of prevalent lower-extremity peripheral arterial disease in the National Health and Nutrition Examination Survey from 1999 to 2004 [14], with cardiovascular disease in men in the Framingham Offspring Study [13], and with a lower hazard ratio of ischemia stroke in men in the Prospective Study of Korean Men and Women [15], respectively. We also used the total bilirubin levels in our study and found that higher total bilirubin levels were associated with lower CV events and all-cause mortality in the HD patients. Fukui et al [42] also reported that lower serum bilirubin was an independent factor for increased CVD risk in hemodialysis patients with type II diabetes. Do Sameiro-Faria et al. [43] suggested that higher bilirubin levels are associated with beneficial effects in HD patients by improving the lipid profile (ox-LDL, Apo A, and Apo B) and reducing inflammatory status (adiponectin, paraoxonase 1, and interleukin-6). In a rat model of adenine-induced kidney failure, Boon et al. [44] discovered that endogenously elevated bilirubin specifically prevented protein and lipid oxidation within the circulatory compartment during kidney inflammation. This study provided further insight into ESRD patients with elevated bilirubin concentrations protecting from vascular injury.

Apart from oxidative stress, vascular calcification is a common complication of chronic kidney disease and is strongly associated with CVD and all-cause mortality in ESRD patients [45,46]. Tanaka et al. [47] demonstrated an inverse relationship between bilirubin concentrations and coronary artery calcification (CAC) measured by noninvasive multi-slice computed tomography. Sung et al. [48] also showed an inverse relationship between total serum bilirubin and CAC score in healthy men. These data indicate that ESRD patients with elevated bilirubin concentrations are protected from coronary artery vascular calcification and thus may explain the low prevalence of CVD events and mortality in these patients. Bilirubin might protect from endothelial dysfunction, vascular calcification, hypertension, lipoprotein oxidation/ and dyslipidemia in ESRD patients undergoing HD and, subsequently, might contribute to low CVD events and mortality [49].

This study has some limitations. First, the study patients were prevalent patients instead of incident patients. As the median HD vintage was 52 months, study patients might have represented selected survivors. However, the distributions of genotypes of *HO-1* and *UGT1A1* in this cohort were similar to those in the healthy controls and showed no difference between censored and non-censored patients or among patients with different dialysis vintages. Thus, the possibility of survivor bias might be low. Second, the findings of our study are applicable to the Taiwanese population and confirmatory replication studies in other populations are needed.

## 5. Conclusions

In summary, our cohort study shows that the polymorphisms of the *HO-1* and *UGT1A1* genes cooperate to affect bilirubin levels and might have an important impact on preventing the development of CVD and death among chronic HD patients. Accurate risk stratification that takes into account serum bilirubin and genetic information would allow recognition of high-risk groups. Molecules that either induce HO activity or partially inhibit UGT1A1 could be used therapeutically to induce the protective effects of bilirubin in such high-risk populations.

## Figures and Tables

**Figure 1 antioxidants-10-01403-f001:**
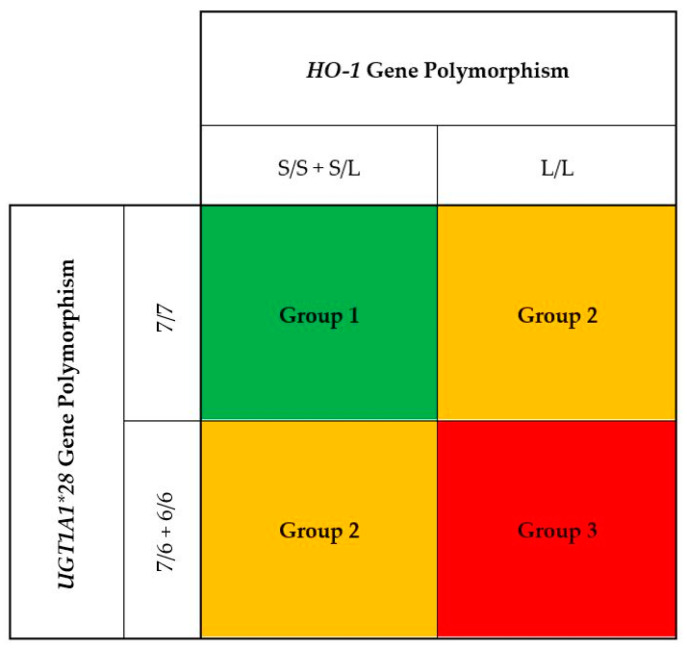
A total of 1080 hemodialysis patients were classified into three groups by the combination of three genotypes of the *HO-1* gene promoter (S/S, S/L, and L/L) with three genotypes of the *UGT1A1* gene (7/7, 7/6, and 6/6). We constructed three genetic models including S/S + S/L with 7/7 (Group 1); L/L with 7/7 and S/S + S/L with 7/6 + 6/6 (Group 2); and L/L with 7/6 + 6/6 (Group 3).

**Figure 2 antioxidants-10-01403-f002:**
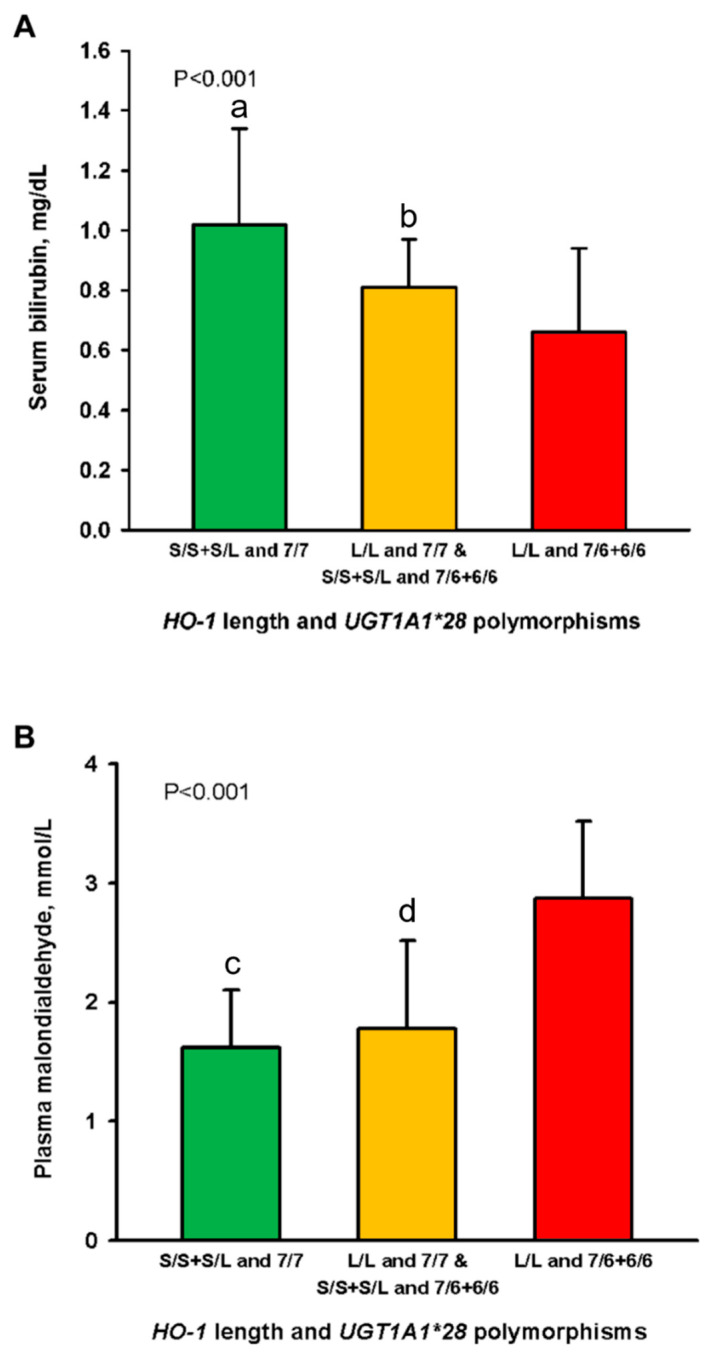
Levels of serum bilirubin (**A**) and plasma malondialdehyde (**B**) in 1080 hemodialysis patients stratified into three groups by the combination of promotor polymorphisms of the *HO-1* and *UGT1A1* genes. Abbreviations: HO, heme oxygenase; *UGT1A1*, uridine diphosphate–glucuronosyl transferase. *p* shows trends <0.001 among the three groups in Figure 2A,B. ^a,c^
*p* < 0.01 and ^b,d^
*p* < 0.05 as compared to the patients of the L/L and 7/6 + 6/6 genotypes.

**Table 1 antioxidants-10-01403-t001:** Baseline demographic and laboratory characteristics of hemodialysis patients stratified by genotype combination of *HO-1* and *UGT1A1* polymorphism.

	Combined Genotypes of *HO-1* and *UGT1A1* Polymorphisms			
Parameters	Group 1 (*n* = 37)	Group 2 (*n* = 729)	Group 3 (*n* = 314)	*p* Value
Age, years	59 ± 15	58 ± 14	60 ± 13	0.067
Men, n (%)	15 (40.5)	365 (50.1)	172 (54.8)	0.160
Current smoker, n (%)	13 (35.1)	232 (31.8)	74 (23.6)	0.076
Hypertension, n (%)	15 (40.5)	413 (56.7)	192 (61.1)	0.053
Diabetes mellitus, n (%)	10 (27.0)	223 (30.6)	105 (33.4)	0.562
Previous CVD disease, n (%)	9 (24.3)	210 (28.8)	98 (31.2)	0.583
HD duration, months	50 ± 32	49 ± 11	51 ± 25	0.471
Body mass index, kg/m^2^	22.3 ± 2.9	21.8 ± 3.3	22.1 ± 3.1	0.513
Systolic BP, mmHg	137 ± 22	139 ± 23	138 ± 22	0.665
Diastolic BP, mmHg	77 ± 11	78 ± 11	77 ± 10	0.832
Total cholesterol, mg/dL	159 ± 39	172 ± 35	174 ± 41	0.542
Triglyceride, mg/dL	114 ± 46	166 ± 120	156 ± 103	0.417
LDL-cholesterol, mg/dL	100 ± 21	111 ± 27	115 ± 34	0.378
ALT, U/L	22 ± 2	25 ± 4	20 ± 3	0.567
AST, U/L	24 ± 5	27 ± 5	21 ± 4	0.612
GGT, U/L	29 ± 7	31 ± 4	30 ± 5	0.705
Albumin, g/dL	3.9 ± 0.3	3.9 ± 0.3	3.9 ± 0.4	0.973
Hs-CRP, mg/L	5.60 (1.33, 7.01)	4.73 (1.27, 5.15)	5.44 (1.51, 6.93)	0.108
Hemoglobin, g/dL	10.4 ± 1.9	10.3 ± 1.5	10.3 ± 1.5	0.912
Ferritin, μg/L	369 (204, 497)	362 (206, 624)	351 (197, 607)	0.614
Transferrin saturation, %	28 ± 14	33 ± 16	33 ± 15	0.177

ALT, alanine aminotransferase; AST, aspartate aminotransferase; BP, blood pressure; CVD, cardiovascular disease; GGT, gamma-glutamyl transferase; HD, hemodialysis; HO, heme oxygenase; Hs-CRP, high-sensitivity C-reactive protein; LDL, low-density lipoprotein; *UGT1A1*, uridine diphosphate–glucuronosyl transferase 1A1. The combinations of the six genotypes, S/S, S/L, and L/L of the *HO-1* gene and 7/7, 7/6, and 6/6 of the *UGT1A1* gene, were S/S + S/L and 7/7 in Group 1, L/L and 7/7 and S/S + S/L and 7/7 + 6/6 in Group 2; and L/L and 7/6 + 6/6 in Group 3, respectively.

**Table 2 antioxidants-10-01403-t002:** Adjusted hazard ratio (95% confidence interval) of tertiles of bilirubin and various combinations of promotor polymorphisms of the *HO-1* and *UGT1A1* genes in chronic hemodialysis patients.

	Genotype Group 1 vs. Group 3 (Reference)	Genotype Group 2 vs. Group 3 (Reference)
	**Adjusted HR (95% CI)**	**Adjusted HR (95% CI)**
**Cardiovascular events**	0.35 (0.15−0.89) *p* = 0.024	0.63 (0.50−0.83) *p* < 0.001
**All-cause mortality**	0.40 (0.16−0.99) *p* = 0.049	0.81 (0.68−1.01) *p* = 0.060
	**Bilirubin** Middle tertile vs. lower tertile (reference)	**Bilirubin** Upper tertile vs. lower tertile (reference)
	**Adjusted HR (95% CI)**	**Adjusted HR (95% CI)**
**Cardiovascular events**	0.72 (0.50−0.96) *p* = 0.045	0.40 (0.30−0.53) *p* < 0.001
**All-cause mortality**	0.90 (0.59−1.01) *p* = 0.079	0.57 (0.41−0.98) *p* = 0.023

A multivariate Cox regression model was adjusted for age, sex, smoking status, diabetes, prior cardiovascular disease, body mass index, total cholesterol, systolic blood pressure, hemodialysis. duration, urea Kt/V, serum albumin, high-sensitivity C-reactive protein, hemoglobin, and plasma malondialdehyde. Abbreviations: HR, hazards ratio, CI, confidence interval. The combinations of the six genotypes, S/S, S/L, and L/L of the *HO-1* gene and 7/7, 7/6, and 6/6 of the *UGT1A1* gene, were S/S + S/L and 7/7 in Group 1; L/L and 7/7 and S/S + S/L and 7/7 + 6/6 in Group 2; and L/L and 7/6 + 6/6 in Group 3, respectively.

## Data Availability

All data generated or analyzed during this study are included in this published article.

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
