# Peer review of "Bilirubin Links HO-1 and UGT1A1*28 Gene Polymorphisms to Predict Cardiovascular Outcome in Patients Receiving Maintenance Hemodialysis"

_antioxidants, 2021, doi:10.3390/antiox10091403_

Round 1
Reviewer 1 Report
The manuscript submitted deals with HO-1 and UGT1A1*28 polymorphisms and cardiovascular outcomes in hemodialysis patients.
The assessment of this research question was performed by combining the polymorphisms thereby creating different groups and assessing CVD events and all cause mortality.
The design was appropriate and the rational for performing this investigation appropriately described.
However, some points in the MS can be improved or needs verification:
1.) The authors defined 3 groups based on the genotypes. For me this grouping is not clear and needs a better description why this was performed and why the statistical evaluation was based on these groups.
2.) Mild hyperbilirubinaemia is phenotypically defined by increased bilirubin levels but without having increased liver enzymes. Although this MS is based on genotyping it is needed to show the liver enzyme data in addition to the bilirubin levels. Particularly in patients the bilirubin levels could be increased due to liver disease. Since you are presenting bilirubin levels in the paper the enzymes must be accompanied.
3.) You are discussing unconjugated bilirubin but present total bilirubin. For this paper the UCB levels would be of interest.
4.) The statistics could be improved regarding sex and age. You should include at least the differentiation regarding sex. Age would also be of interest, depending on the age range of your group.
Minor points:
There are some typos in the MS or issues with the use of the wrong time. e.g. lines 38 or 57
Figure 2: the p value is significantly different to ?
All cause mortality: Why is cancer missing?
line 264: Boon instead of Bonn
Reviewer 2 Report
Paper by Ho et al. is a short, well-written report, based on clinical data, analyzing the combined effect of polymorphisms in the promoters of two genes involved in heme and bilirubin metabolism (HMOX1 and UGT1A1) in hemodialyzed patients. The manuscript provides an interesting confirmation of the significance of HO-1 as a protective protein, indicating the role of bilirubin, the direct derivative of biliverdin, the HO-1 product. The Authors found that carriers of the more active HMOX1 promoter alleles and less active UGT1A1 variant have a higher level of bilirubin, reduced oxidative stress, and lower risk of cardiovascular events. The work is based on correlation analyzes, without examining the biological mechanisms of the observed relationships. However, relatively high number of patients enrolled, and relatively long follow-up period increase the value of the results.
Minor concerns:
- The Authors use n=27 of GT repeats as a threshold value to classify HMOX1 promoter allele as short (S) or long (L) variant. Please, discuss shortly the biological rationale for this threshold. It would be beneficial to refer studies directly analyzing the effect of GT repeat numbers and significance of the presence of one or two S alleles on the HMOX1 expression level (e.g. Taha et al. Arterioscler Thromb Vasc Biol 2010; 30:1634-1641).
- Group 1 contains relatively lower proportion of patients with hypertension. Please, comment the possible reason/consequence of that.
- The Authors analyze the all-cause mortality. Is there any association between the HMOX1/UGT1A1 polymorphisms and CVD-related mortality? Please, add this information.
Typos/editorial errors:
- Is this the correct version of the title? Maybe “to predict” would be better?
- Abbreviations (e.g. LDL, BP) should be explained on first use in the text.
- Page 2, line 55: maybe “induction” instead of “induce” and “inhibition” instead of “inhibit” would be better?
- Page 8, line 248: maybe “attenuate inflammation” instead of “contrast inflammation” would be better?
Round 2
Reviewer 1 Report
The comments have been addressed.
I recommend to also include gamma-GT in table 1 to the other liver enzymes.
Author Response
Dear Reviewer,
Thank you so much for the comments and recommendations to our submitted manuscript. We have added gamma-GT, in Table 1, and there were no significant differences in gamma-GT levels among the three groups (Line 19 of Table 1, Page 5).